# Impacts of Structure-Directing Agents on the Synthesis of Cu₃Mo₂O₉ for Flexible Lignin-Based Supercapacitor Electrodes

**Siddhi Mehta** [1], **Sangeetha Kumaravel** [2,3], **Swarn Jha** [4], **Matthew Yen** [5], **Subrata Kundu** [2,3] **and Hong Liang** [1,4,*]

1   Department of Materials Science and Engineering, Texas A&M University, 575 Ross Street, College Station, TX 77843-3127, USA
2   Electrochemical Process Engineering (EPE) Division, CSIR-Central Electrochemical Research Institute (CECRI), Karaikudi 630003, India
3   Academy of Scientific and Innovative Research (AcSIR), Ghaziabad 201002, India
4   J. Mike Walker '66 Department of Mechanical Engineering, Texas A&M University, 202 Spence Street, College Station, TX 77843-3123, USA
5   Artie McFerrin Department of Chemical Engineering, Texas A&M University, College Station, TX 77843-3123, USA
*   Correspondence: hliang@tamu.edu; Tel.: +1-(979)-862-2623; Fax: +1-979-845-3081

**Abstract:** Due to demands for sustainability, the interest in energy storage devices constructed from green materials has increased immensely. These devices currently have yet to be satisfactory. Issues include high production costs and toxicity, limited dependability, and subpar electrochemical performance. In this research, low-cost, plant-based electroactive Cu₃Mo₂O₉ materials were synthesized via co-precipitation followed by an annealing method using two different structure-directing agents, i.e., the commonly used surfactant cetyltrimethylammonium bromide (CTAB) and the biomolecule deoxyribonucleic acid (DNA) as a greener alternative, and these materials were studied for the first time. Further, the Cu₃Mo₂O₉ nanoparticles developed using CTAB and DNA were integrated into the lignin matrix and studied as flexible electrodes for supercapacitor application. Here, the morphological advantages of the nanorods and nanosheets formed by varying the synthesis methods and their effects during supercapacitor studies were studied in detail. After 1200 cycles, the Al/lig–Cu₃Mo₂O₉@DNA supercapacitor exhibited higher capacitive performance when compared to the Al/lig–Cu₃Mo₂O₉@CTAB supercapacitor. The Al/Lig-Cu₃Mo₂O₉@DNA supercapacitor had an initial specific capacitance of 404.64 mF g⁻¹ with a ~70% retention, while the Al/Lig-Cu₃Mo₂O₉@CTAB supercapacitor had an initial specific capacitance of 309.59 mF g⁻¹ with a ~50% retention. This study offers a new approach to creating scalable, low-cost, green composite CuMoO₄-based electrodes for flexible supercapacitors.

**Keywords:** metal molybdates; supercapacitors; DNA; CTAB; lignin; nanomaterials; biomaterials; composites; electrodes; clean energy storage

## 1. Introduction

Engineered nanomaterials have undergone substantial study and development in recent years owing to their many potential applications. Nanoparticles exhibit unique physiochemical properties that make them more effective compared to bulk materials [1]. Engineered nanoparticles have mainly been used in electronics, biotechnology, and energy storage applications [2]. Nanoparticle synthesis typically involves toxic and expensive chemicals that can have a negative effect on the environment, which is a significant problem for large-scale synthesis. This emerging issue of sustainability has given green chemistry much attention due to its cost-effective, safe, and energy-efficient nature [3]. Nanoparticles synthesized through green routes mainly involve phytochemicals and organic acids, resulting in ecofriendly products [4].

Lignin is an abundant biomass material produced by the paper and pulp industries as byproducts [5]. The porous structure of lignin-derived carbonaceous materials makes them suitable electrode materials for supercapacitors and other energy storage devices [6–15]. The exceptional specific surface area, active sites, porosity, and electrical conductivity of lignin-derived carbonaceous nanoparticles make them especially promising for developing green energy storage systems [16]. Park et al. used lignin to synthesize a hydrogel electrolyte and free-standing nanofiber electrodes, demonstrating mechanical robustness, high ionic conductivity, and an interconnected porous structure allowing for excellent charge storage [17]. In our previous study, a lignin-based supercapacitor decorated with $MnO_2$ ions to enhance charge transport was synthesized. The lignin-based electrode had high porosity with a 3D morphology and high surface packing density, enabling high areal-specific capacitance, power density, energy retention, and density [12]. However, the complex chemical structure of lignin, pore size distribution, electrolyte systems, and morphology are all characteristics that are not clearly understood in the optimization of lignin-based electrodes [18,19]. As a way to provide pseudocapacitance and boost the electrochemical storage performance of a lignin-based supercapacitor, lignin has been integrated with conductive polymers, transition metal oxides, as well as other carbon-based materials to manufacture composite electrodes [20]. Transition metal oxides (TMOs), such as NiO, $MnO_2$, $RuO_2$, and $Co_2O_3$, have successfully been integrated with lignin, resulting in high-performance supercapacitors [6,12,21].

The study of TMOs has recently been expanded to include binary and ternary components, which exhibit superior electronic structures and better electrochemical characteristics. The multiple metal complexes in TMOs have advantages over their single metal oxide counterparts in terms of higher electrical conductivity and enhanced electrochemical activity [22]. As a result, recent research is concentrated on the usage of metal molybdates ($MnMoO_4$, $NiMoO_4$, $CoMoO_4$, etc.) to augment the capacitive performance of the electroactive materials in supercapacitors [23,24]. Binary metal oxides, as well as metal molybdates, have recently proven to be exceptional supercapacitor electrode materials, attracting a lot of attention to this topic. Numerous metal molybdates, namely $MnMoO_4$, $NiMoO_4$, $Bi_2MoO_6$, $CoMoO_4$, $ZnMoO_4$, and $CuMoO_4$, have been successfully synthesized and illustrated their electrochemical capacitive properties [25–30]. Metal molybdates have been established as good electroactive materials resulting from the combined effect of the metal and molybdate species. Metal molybdates have garnered significant intrigue as electroactive materials in supercapacitors due to their cost-effective, environmentally-sound nature, oxidation-reduction properties, stable structure of their crystals, and high electrical conductivity. It has been implemented in various applications, like luminescence, catalysis, and energy storage [31]. $CuMoO_4$ is considered the most valuable among metal molybdates because of its cost-effectiveness, abundance, superb redox nature, and high performance [32]. Different methods, including the hydrothermal route, carbothermal reaction, sol-gel method, microwave synthesis, coprecipitation method, and chemical route synthesis, have been used by researchers to create binary metal molybdates [33]. In one study, a hydrothermal route was implemented to create nanocone arrays of $CuMoO_4$ on Ni foam [34]. For a pouch-type supercapacitor design, a large specific capacitance of 609.7 mF cm$^{-2}$, and high power and energy densities of 2.73 mW cm$^{-2}$ and 0.21 mWh cm$^{-2}$, respectively, were observed. In another study, the fabricated $CuMoO_4$ nanoparticles showed a specific capacitance of 127 F g$^{-1}$ [29]. The nanopowders of $CuMoO_4$ have also been used successfully as an electrode material. The supercapacitor displayed a very high specific capacitance of 210 F g$^{-1}$ (scan rate = 10 mV s$^{-1}$) [35]. Despite its high-performance behavior, developing $CuMoO_4$-based electrodes with the required electrochemical properties for supercapacitors poses a challenge since copper molybdate has not yet undergone a comprehensive investigation as an electrode material [36,37].

In this study, we report solid-state, lightweight, low-cost, $Cu_3Mo_2O_9$ nanoparticles that have been synthesized and integrated into the lignin matrix to fabricate flexible supercapacitors for applications in flexible electronics. The $Cu_3Mo_2O_9$ nanoparticles and

lignin exert a co-operative effect resulting in higher specific capacitances compared to the conventional electric double-layer capacitor (EDLC) on account of enhanced interfacial Faradic reactions. The $Cu_3Mo_2O_9$ nanoparticles were produced by coprecipitation followed by an annealing method using two different structure-directing agents, i.e., the commonly used surfactant cetyltrimethylammonium bromide (CTAB) and the biomolecule deoxyribonucleic acid (DNA) as a greener alternative. Two asymmetric flexible supercapacitors were assembled using a NaOH electrolyte. The supercapacitor performance was evaluated using electrochemical tests, namely, electrochemical impedance spectroscopy (EIS), cyclic charge–discharge (CCD), and cyclic voltammetry (CV). The initial specific capacitance for Al/Lig-$Cu_3Mo_2O_9$@CTAB is 309.59 mF $g^{-1}$, and for Al/Lig-$Cu_3Mo_2O_9$@DNA is 404.64 mF $g^{-1}$. After 1200 cycles, the Al/Lig-$Cu_3Mo_2O_9$@DNA exhibits a comparatively higher retention performance of ~70%, while Al/Lig-$Cu_3Mo_2O_9$@CTAB has a retention of ~50%. The favorable performance of the Al/Lig-$Cu_3Mo_2O_9$@DNA supercapacitor makes it suitable for various energy-storage applications. This work could be a significant step towards the advance of green technology because of the straightforward, affordable procedures employed and the raw materials acquired from environmentally friendly sources.

## 2. Materials and Methods

### 2.1. Materials

The precursor, Copper (II) nitrate trihydrate ($Cu(NO_3)_2$.3$H_2O$, sodium molybdate ($Na_2MoO_4$) dehydrate, cetyltrimethylammonium bromide (CTAB), the biomaterial activated carbon (AC), and the biomolecule ds deoxyribonucleic acid (double-stranded DNA), with a base pair of 50k, were received from Sigma Aldrich. Deionized water (DI) was used in the synthesis and characterization studies of $Cu_3Mo_2O_9$@DNA and $Cu_3Mo_2O_9$@CTAB. The alkaline lignin powder (lig) was purchased and used as received from VWR. The binder Polyvinylidene fluoride (PVDF) and solvent N-Methylpyrrolidone (NMP) were also purchased from Sigma Aldrich. All reagents and chemicals were used as obtained from Sigma-Aldrich and VWR without any further purifications and modifications.

### 2.2. Preparation and Characterization of $Cu_3Mo_2O_9$@DNA and $Cu_3Mo_2O_9$@CTAB

The synthesis of $Cu_3Mo_2O_9$@DNA and $Cu_3Mo_2O_9$@CTAB was carried out by coprecipitation followed by an annealing method using two different structure-directing agents, i.e., CTAB and DNA. The synthesis is represented schematically in Figure 1.

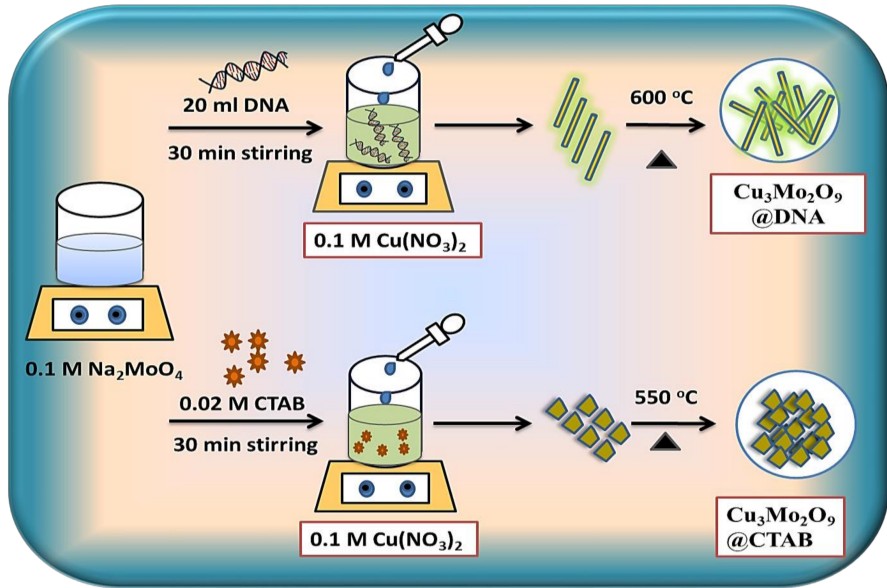

**Figure 1.** Schematic of $Cu_3Mo_2O_9$@DNA and $Cu_3Mo_2O_9$@CTAB synthesis by coprecipitation followed by annealing method.

For the preparation of $Cu_3Mo_2O_9$ developed over DNA ($Cu_3Mo_2O_9$@DNA), the stock solution of DNA was initially prepared by adding 0.12 M DNA powder to 100 mL of DI water and kept stirring for 12 h [38,39]. The dispersion process of DNA powder in the DI water was achieved with the continuous stirring process. 20 mL of the clear stock solution of DNA was added to 60 mL of 0.1 M $Na_2MoO_4.2H_2O$ solution and kept stirring for 30 min. Here, the positively charged metal ions ($M^+$) and the negatively charged groups of DNA, such as hydroxyl, amides, and sugar moieties, interacted electrostatically. In a separate beaker, 50 mL of 0.1 M $Cu(NO_3)_2.3H_2O$ was prepared in DI water, added drop-wise to the DNA-metal ion solution, and constantly stirred for 30 min. After the addition of $Cu(NO_3)_2.3H_2O$ to the above solution, the color changed to pale green and formed the precipitate. The formed precipitate was kept to dry for 20 h at 60 °C. The dried pale green sample was then annealed for 2 h at a temperature of 600 °C. The formed $Cu_3Mo_2O_9$@DNA displayed a bright green color.

Similarly, the $Cu_3Mo_2O_9$ developed over CTAB ($Cu_3Mo_2O_9$@CTAB) was prepared by using the surfactant cetyltrimethylammonium bromide (CTAB). Here, the 0.02 M CTAB powder was directly added to 60 mL of 0.1 M $Na_2MoO_4.2H_2O$ and constantly stirred for 30 min. The 0.1 M $(Cu(NO_3)_2.3H_2O$ solution prepared in 50 mL DI water was then added to the above mixture at a temperature of 60 °C and constantly stirred. The pale green precipitate formed was kept to dry at 60 °C for 20 h. The dried sample was annealed at 550 °C and formed $Cu_3Mo_2O_9$@CTAB. Both samples ($Cu_3Mo_2O_9$@DNA and $Cu_3Mo_2O_9$@CTAB) were then further characterized.

### 2.3. Fabrication of Al/lig–$Cu_3Mo_2O_9$ Composite Electrodes

Two separate mixtures were made by adding the $Cu_3Mo_2O_9$@DNA and $Cu_3Mo_2O_9$@CTAB nanoparticles to lignin (alkaline, TCI). The binder solution of PVDF and NMP was added to the earlier obtained mixture to form two separate slurries. The ratio selected was 10:80:10 indicates 10% lignin, 80% $Cu_3Mo_2O_9$, and 10% PVDF by wt.%. The slurries were coated (mass loading = 4.5 mg cm$^{-2}$) on two separate aluminum (Al) foil substrates (thickness = 0.98 mm) of 4 cm diameter with adjoining strips of 1 cm × 3 cm. The electrodes were heated at 100 °C in a vacuum atmosphere for four hours to complete the fabrication of two separate Al/lig–$Cu_3Mo_2O_9$ composite electrodes.

### 2.4. Asymmetric Solid-State Supercapacitor Assembly

The previously made composite electrodes were used to assemble the asymmetric supercapacitor, the Al/lig–$Cu_3Mo_2O_9$-based electrode (anode), and Al/AC electrode (cathode). A paper separator coated with NaOH electrolyte was sandwiched between the electrodes to complete the assembly. The supercapacitors were named Al/lig–$Cu_3Mo_2O_9$@CTAB and Al/lig-$Cu_3Mo_2O_9$@DNA, according to the fabrication method of the $Cu_3Mo_2O_9$ nanoparticles.2.5. characterization and measurements.

X-ray diffraction (XRD) was performed via a PAN analytical Advanced Bragg-Brentano X-ray powder diffractometer (XRD) with Cu Kα radiation ($\lambda$ = 0.154178 nm; scan rate = 5° min$^{-1}$; 2θ range = 10–80°). The LASER Raman spectroscopy was carried out with a Renishaw in Via Raman Microscope and an excitation wavelength of 532 nm (He-Ne laser). The Scanning electron microscopic (SEM) study was done with the TESCAN model with a magnification range of 30×–300,000× with an accelerated voltage capability of 0.3–30 kV. SEM equipment with a separate detector was used to conduct the Energy Dispersive X-ray Spectroscopy study (EDS). Using Theta Probe AR-XPS System from Thermo Fisher Scientific, East Grinstead, UK, X-ray photoelectron spectroscopy (XPS) was used to evaluate the materials and determine the oxidation state of the elements present. A Gamry potentiostat (version 6.33 from Gamry Instruments, Warminster, PA, USA) was used for all electrochemical characterizations.

## 3. Results and Discussion

### 3.1. X-ray Diffraction (XRD) Spectra and Raman Spectroscopy

XRD analysis was carried out for both samples, and the diffraction patterns were stacked in Figure 2a. In both the XRD patterns, the presence of crystalline planes (corresponding diffraction peaks marked with red asterisks (*)), such as (002), (101), (011), (102), (103), (013), (200), (201), (004), (113), (202), (210), (104), (022), (121), (203), (122), (105), (204), (220), (024), (214), (124), (223), (224), (133), (230), (207), (217), (040), and (128), are observed with their corresponding diffraction angles for $Cu_3Mo_2O_9$ [40,41]. These observed diffraction patterns perfectly matched with the JCPDS file no. 010-6455, confirming the formation of orthorhombic $Cu_3Mo_2O_9$ developed over DNA and CTAB.

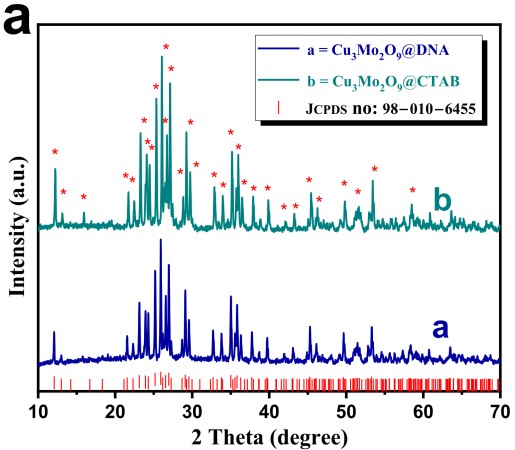
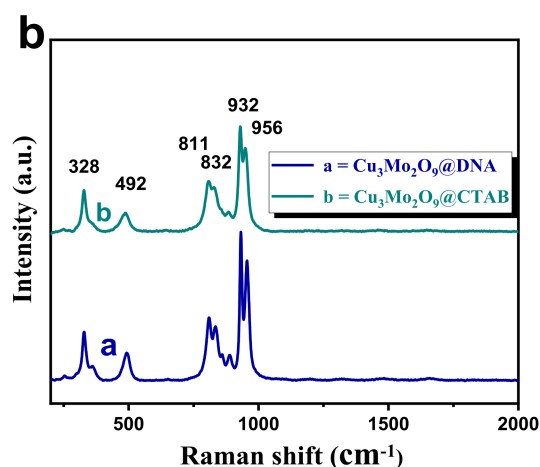

**Figure 2.** (**a**) Stacked XRD (diffraction peaks marked with red asterisks (*)) of both $Cu_3Mo_2O_9$@DNA and $Cu_3Mo_2O_9$@CTAB; (**b**) Raman analysis of both $Cu_3Mo_2O_9$@DNA and $Cu_3Mo_2O_9$@CTAB.

Further, the vibrational property of $Cu_3Mo_2O_9$ was measured using Raman spectroscopy, and the results are given in Figure 2b. Here, the stacked spectra of $Cu_3Mo_2O_9$@DNA and $Cu_3Mo_2O_9$@CTAB show the presence of a vibrational peak at 328, 492, 811, 832, 932, and 956 cm$^{-1}$, which confirms the formation of the Raman active modes of the sample [35,42]. The peak at 956 cm$^{-1}$ matches the vibration along the zig-zag nature of Mo-O bonds in $Cu_3Mo_2O_9$. Additionally, the 932 cm$^{-1}$ is ascribed to the symmetric stretching vibration of Mo-O bonds, and the peaks at 811 and 832 cm$^{-1}$ correspond to the presence of the stretching mode of the linked Mo–O–Mo bonds. The lower vibrational peaks, such as 328 and 492 cm$^{-1}$, correspond to the presence of Cu-O in $Cu_3Mo_2O_9$. The absence of the D band and G band for the denatured structure directing agents (DNA and CTAB) after annealing indicates that there is no decomposition or partial carbonization. A similar kind of Raman spectra has been noted for annealed $CuMoO_4$ material, as reported in [43]. Thus, from the observed XRD and Raman analysis, the $Cu_3Mo_2O_9$ sample developed using DNA and CTAB was confirmed.

### 3.2. Scanning Electron Microscopy (SEM)

In order to study the morphological analysis of $Cu_3Mo_2O_9$@DNA and $Cu_3Mo_2O_9$@CTAB, SEM analysis was performed and is shown in Figure 3. From the low-magnification SEM images of $Cu_3Mo_2O_9$@DNA (Figure 3a), a nanorod structure can be observed with a ~1.2 µm length and ~400 nm diameter. The individual nanorods of $Cu_3Mo_2O_9$ are clearly visible in the SEM images under high magnification (Figure 3b). However, in the SEM images of $Cu_3Mo_2O_9$@CTAB under low as well as high magnification (Figure 3c,d), perfect nanosheets between 150–200 nm in thickness are observed.

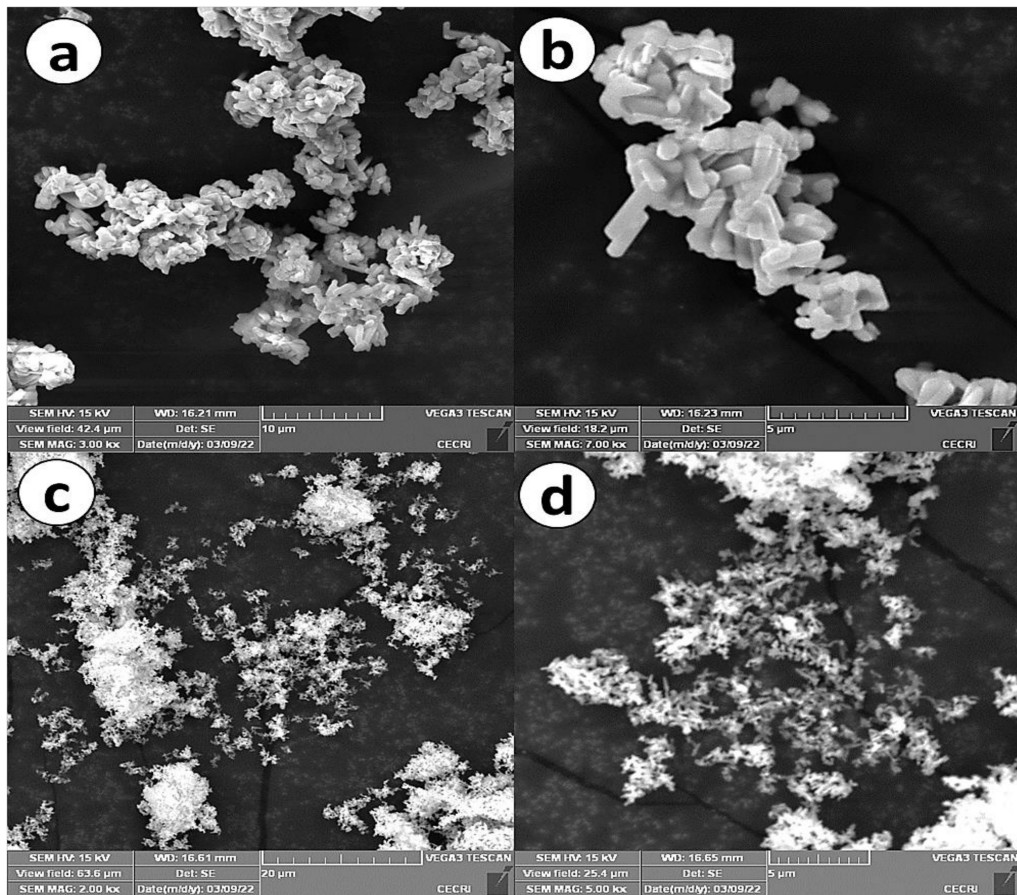

**Figure 3.** (**a**) SEM images (low magnification) of Cu₃Mo₂O₉@DNA; (**b**) SEM images (high magnification) of Cu₃Mo₂O₉@DNA; (**c**) SEM images (low magnification) of Cu₃Mo₂O₉@CTAB; (**d**) SEM images (high magnification) of Cu₃Mo₂O₉@ CTAB.

### 3.3. Energy-Dispersive X-ray Spectroscopy (EDS)

The elemental presence in both samples was confirmed with EDS analysis obtained from the separated detector used in the SEM instrument (Figure 4a,b). For both samples, elements such as Cu, O, and Mo were present. Additionally, the presence of carbon, which is obtained from the DNA and CTAB used to synthesize the samples, was observed. More precisely, for both the EDS spectra, the extra peaks for any impurities were not identified, suggesting the pure form of the $Cu_3Mo_2O_9$ samples.

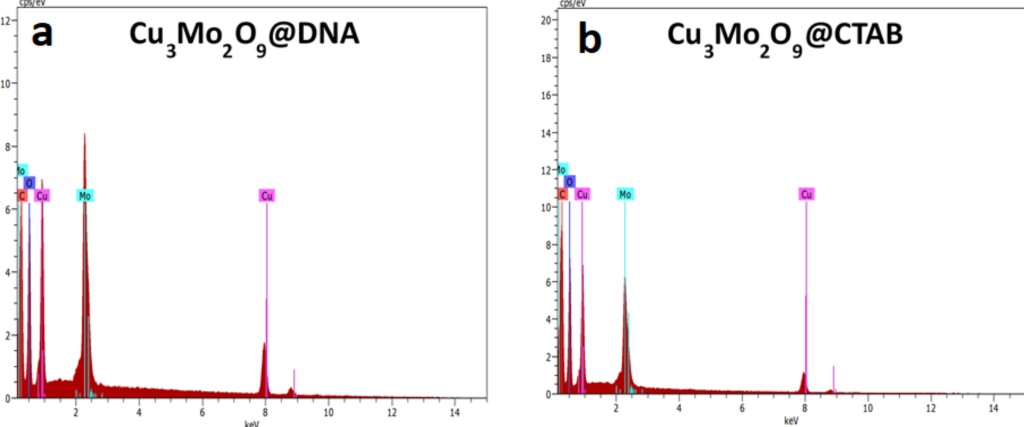

**Figure 4.** EDS spectra of (**a**) Cu₃Mo₂O₉@DNA; (**b**) Cu₃Mo₂O₉@CTAB.

*3.4. X-ray Photoelectron Spectroscopy (XPS)*

The oxidation state of the sample was studied in detail via XPS for both the $Cu_3Mo_2O_9$@DNA and $Cu_3Mo_2O_9$@CTAB samples (Figure 5). The elemental presence, such as Cu 2p, Mo 3d, O 1s, N1s, and C1s peaks, with their corresponding binding energy values, was noted in both samples, showing the formation of $Cu_3Mo_2O_9$ from DNA and CTAB, respectively (Figure 5a,e) [44]. In the high-resolution XPS spectra of Cu 2p, the presence of Cu $2p_{1/2}$ and $2p_{3/2}$ formed during the spin-orbital splitting, with binding energy values such as 932.9 and 960.5 eV, corresponding to the presence of $Cu^{2+}$ in $Cu_3Mo_2O_9$@DNA (Figure 5b). In addition, the presence of satellite peaks was noted with the binding energies 939.7, 941.83, and 960.43 eV for $Cu^{2+}$. Similarly, in the high-resolution spectra of Cu 2p for $Cu_3Mo_2O_9$@CTAB, the presence of $Cu^{2+}$ was noted with their corresponding satellite peak in Figure 5f. The XPS spectra of Mo 3d show the splitting of Mo $3d_{3/2}$ and $3d_{5/2}$ with binding energy values of 233.46 and 230.32 eV, indicating the presence of $Mo^{6+}$ in $Cu_3Mo_2O_9$ (Figure 5c,g). In addition, the high-resolution spectra of the O1s spectra of both $Cu_3Mo_2O_9$ samples showed two peaks at 528.37 and 529.07 eV, corresponding to the M-O bonds. The presence of the C=O peak at 530.1 eV comes from the DNA used for the material synthesis (Figure 5d). Similarly, the presence of the M-O peak and the C-O were noted for the $Cu_3Mo_2O_9$ developed using CTAB (Figure 5h). Thus, the overall observed characterization results suggest that $Cu_3Mo_2O_9$ was formed over the two different structure-directing agents, DNA and CTAB.

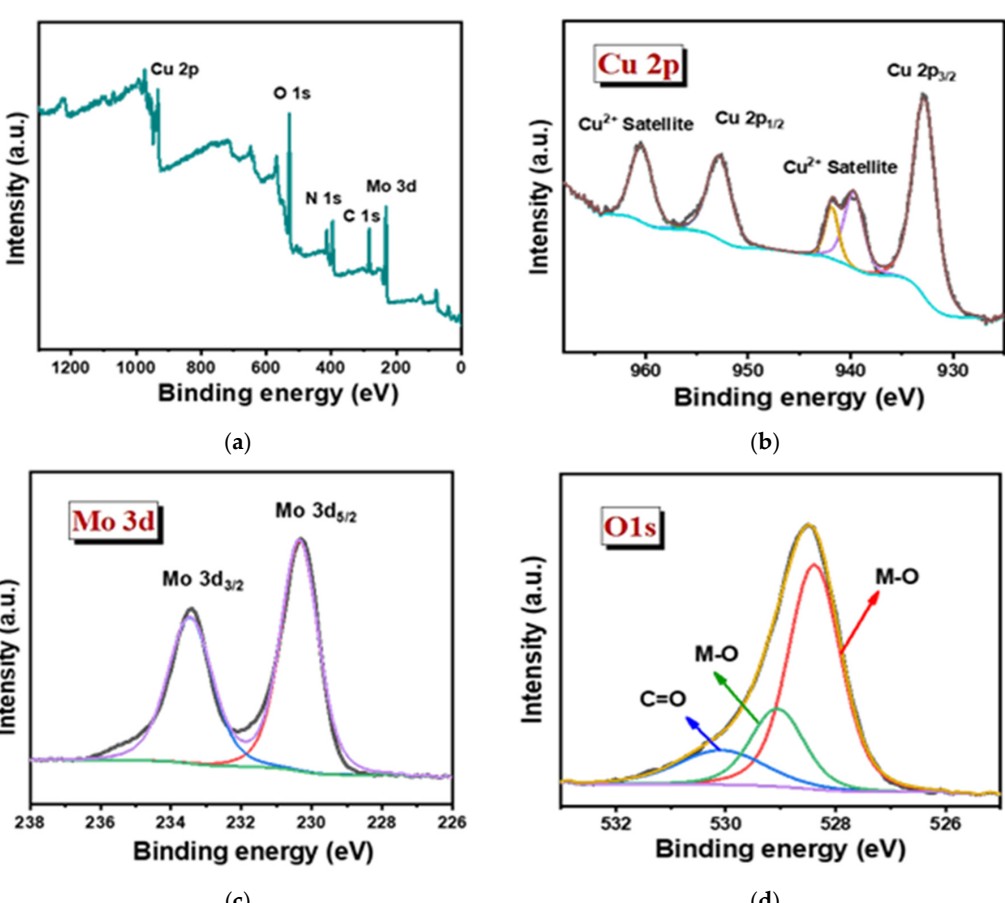

**Figure 5.** *Cont.*

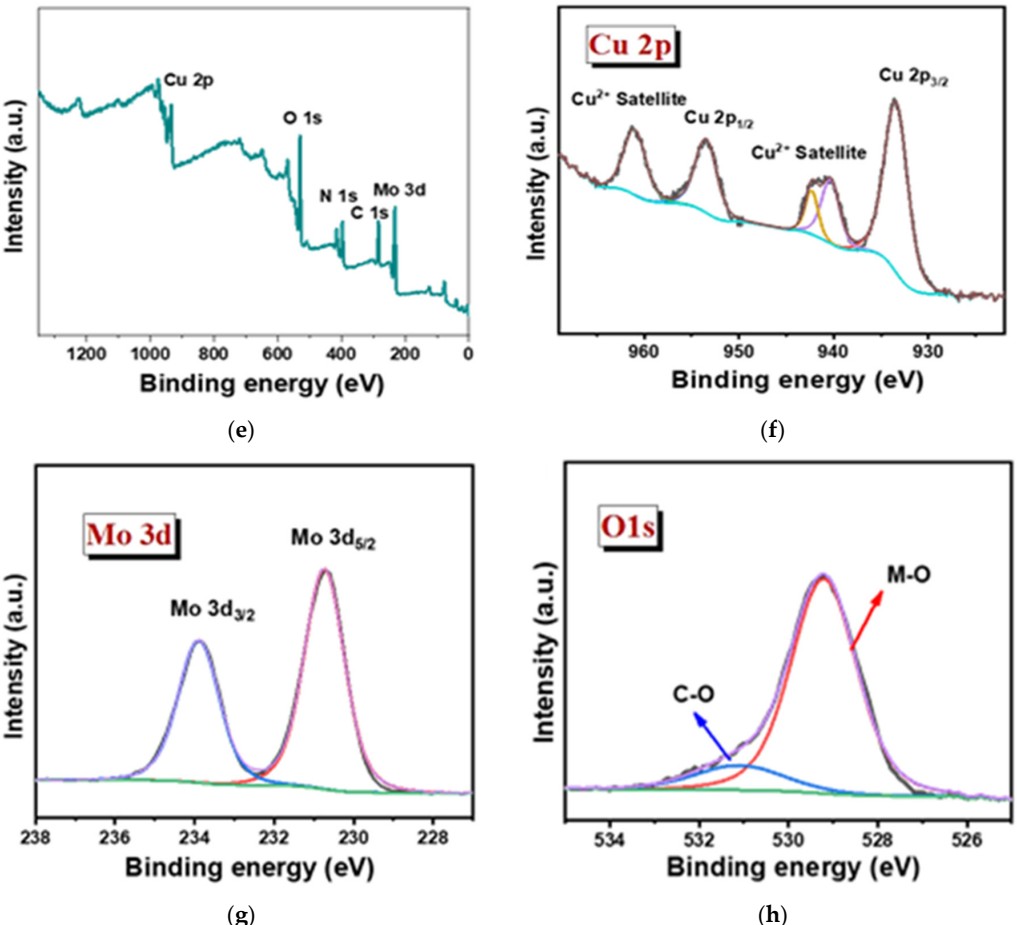

**Figure 5.** (**a**) Survey spectra of Cu$_3$Mo$_2$O$_9$@DNA; (**b**–**d**) high magnified XPS spectra of the individual elements Cu 2p, Mo 3d, and O 1s of Cu$_3$Mo$_2$O$_9$@DNA; (**e**) survey spectra of Cu$_3$Mo$_2$O$_9$@CTAB; (**f**–**h**) highly magnified XPS spectra of the individual elements Cu 2p, Mo 3d, and O 1s of Cu$_3$Mo$_2$O$_9$@CTAB.

Additionally, the presence of N 1s and C1s was obtained from the precursor material used, such as DNA and CTAB. Figure S1a,c show a strong peak for N 1s at 396.1 eV for Cu$_3$Mo$_2$O$_9$@DNA and 396.5 eV for Cu$_3$Mo$_2$O$_9$@CTAB. The high-resolution spectra for C 1s in Cu$_3$Mo$_2$O$_9$@DNA display binding energy values, such as 284.1, 282.87, and 286.58 eV, arising due to the functional groups' presence in DNA, such as with C-O, C(O)-NH, and C=O, which is displayed in Figure S1b. Similarly, the C 1s observed for the sample developed using CTAB shows the presence of C-O, C-C, and C=O at the binding energies, such as 287.21, 284.25, and 283.32 eV, and are displayed in Figure S1d. Thus, the overall observed characterization results suggest the successful formation of Cu$_3$Mo$_2$O$_9$ using the two different structure-directing agents, DNA and CTAB.

### 3.5. Electrochemical Performance

In order to compare and analyze the energy storage performance of the Cu$_3$Mo$_2$O$_9$ nanoparticle-decorated lignin-based asymmetric supercapacitors, electrochemical testing was performed. Initially, cyclic voltammetry (CV) tests were run to study the characteristic behavior of these supercapacitors. In this research, a NaOH electrolyte was used instead of the usually preferred KOH electrolyte. This is due to the quick deterioration of lignin that occurs when it comes into contact with KOH, which results in very poor capacitive performance and a short cycle life. The CV tests were run at multiple scan rates for both supercapacitors (Al/Lig-Cu$_3$Mo$_2$O$_9$@DNA & Al/Lig-Cu$_3$Mo$_2$O$_9$@CTAB) (Figure 6a,b). A paper separator coated with NaOH as the electrolyte was used. The

operating voltage range for both supercapacitors was maintained at 0.5–1.5 V. For higher scan rates, the rate of diffusion increased while the specific capacitance decreased. For slow irreversible chemical reactions, the thickness of the diffusion layer is proportional to the applied potential and decreases at higher applied potentials. As a result, with higher scan speeds, delayed irreversible processes are preferred, resulting in high currents. This eliminates the side reaction peaks and the associated noise from the system, producing a smoother CV curve [12]. This results in a quasi-rectangular form and can be attributed to diffusion restrictions that prevent efficient contact between the electrolyte and the electro-active material [45]. Additionally, a film of impurities is formed on the electrode surface during the charge–discharge process, resulting in a positive reduction current [46]. Hence, a scan rate of 25 mV s$^{-1}$ was adopted for additional CV testing. CV plots for Al/Lig-Cu$_3$Mo$_2$O$_9$@CTAB and Al/Lig-Cu$_3$Mo$_2$O$_9$@DNA are displayed in Figure 6c. A histogram comparing the measured specific capacitance values for the two supercapacitors can be seen in Figure 6d. The maximum specific capacitance value measured for Al/Lig-Cu$_3$Mo$_2$O$_9$@CTAB is 2.5 F g$^{-1}$ (11.2 mF cm$^{-2}$) and for Al/Lig-Cu$_3$Mo$_2$O$_9$@DNA is 3.4 F g$^{-1}$ (17.6 mF cm$^{-2}$). In comparison to the Al/Lig-Cu$_3$Mo$_2$O$_9$@CTAB supercapacitor, the Al/Lig-Cu$_3$Mo$_2$O$_9$@DNA supercapacitor exhibited a greater specific capacitance during the CV testing. The superior performance of the Al/Lig-Cu$_3$Mo$_2$O$_9$@DNA supercapacitor can be attributed to the different structure-directing agents used to fabricate the Cu$_3$Mo$_2$O$_9$ nanoparticles. Due to the varying formation mechanisms of CTAB and DNA, different Cu$_3$Mo$_2$O$_9$ nanostructures with varying degrees of agglomerations were obtained. The associated formation and charge-storage mechanisms are explored in the next section.

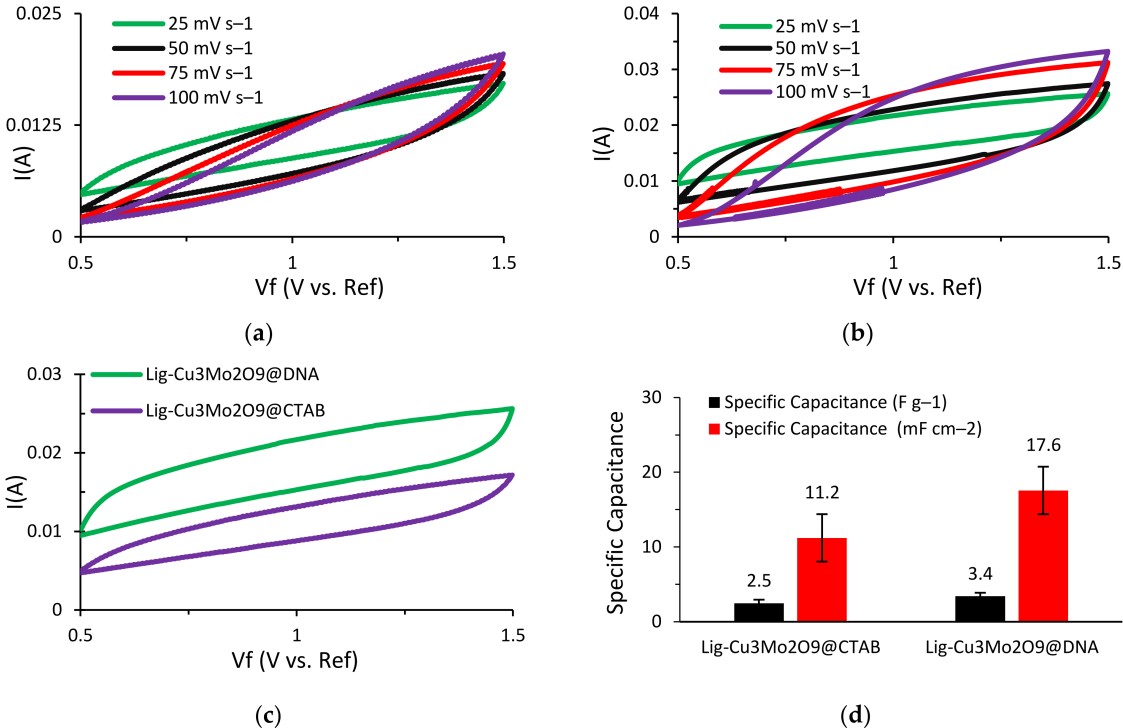

**Figure 6.** CV curves at various scan rates for (**a**) Al/Lig-Cu$_3$Mo$_2$O$_9$@CTAB supercapacitor; (**b**) Al/Lig-Cu$_3$Mo$_2$O$_9$@DNA supercapacitor; (**c**) CV plots of Al/Lig-Cu$_3$Mo$_2$O$_9$@CTAB and Al/Lig-Cu$_3$Mo$_2$O$_9$@DNA at 25 mV s$^{-1}$; (**d**) histogram comparing the specific capacitance values obtained from the CV tests.

The supercapacitor's cycle life and system stability are its two most important criteria. A cyclic charge–discharge (CCD) test was conducted for a total of 1200 cycles in order to examine the capacitance storage capability of the supercapacitors (Figure 7). The specific capacitance performance of the two materials is contrasted in Figure 7a. The initial specific capacitance for Al/Lig-Cu$_3$Mo$_2$O$_9$@CTAB is 309.59 mF g$^{-1}$ (theoretical

capacitance = 20 mF), while the Al/Lig-Cu$_3$Mo$_2$O$_9$@DNA exhibits a 404.64 mF g$^{-1}$ (theoretical capacitance = 26.14 mF) initial specific capacitance. After running for 1200 cycles, the specific capacitance of Al/Lig-Cu$_3$Mo$_2$O$_9$@CTAB dropped down to 74.76 mF g$^{-1}$ (theoretical capacitance = 4.83 mF), and for Al/Lig-Cu$_3$Mo$_2$O$_9$@DNA, dropped down to 280.6 mF g$^{-1}$ (theoretical capacitance = 18.13 mF). In order to provide a proper comparative analysis of both supercapacitors, the y-axis of the cycling stability performance in Figure 7a,b was kept exponential. The capacitance energy retention for both supercapacitors is depicted in Figure 7b. Al/Lig-Cu$_3$Mo$_2$O$_9$@DNA exhibits a comparatively higher retention performance of ~70%, while Al/Lig-Cu$_3$Mo$_2$O$_9$@CTAB displays a retention of ~50%. In terms of cycle life and system stability, the Al/Lig-Cu$_3$Mo$_2$O$_9$@DNA supercapacitor displays better capacitive performance than the Al/Lig-Cu$_3$Mo$_2$O$_9$@CTAB supercapacitor. As a result, it is observed that areal-specific capacitance, as well as capacitance retention, change when the Cu$_3$Mo$_2$O$_9$ nanoparticles synthesis technique is changed. This shows the extraordinary impact of the Cu$_3$Mo$_2$O$_9$@DNA nanoparticles in giving the supercapacitor not only greater capacitance but also performance stability. Figure 7c shows the current density effects on the specific capacitance of both supercapacitors. The specific capacitance drops as the current density is increased from 20 to 150 mA g$^{-1}$ (0.02 to 0.15 A g$^{-1}$). This illustrates a supercapacitor's general functionality. Due to the phenomenon of the regulated mass transfer of the electrolytic ions into the interior section of the electrode material, the obtained specific capacitance is high at a reduced current density [12].

From the Ragone plot obtained in Figure 7d, the correlation between the energy density at various current densities and the resulting power density can be observed. The curve is similar to those seen in supercapacitors made of metal oxides. The energy density exhibits a concave upward drop as opposed to the typical convex upward decline. This behavior of lignin-based supercapacitors can be observed in our previous work [6,12]. For the Al/Lig-Cu$_3$Mo$_2$O$_9$@DNA supercapacitor, the energy and power densities measured at a current density of 20 mA g$^{-1}$ (0.02 A g$^{-1}$) are 40.74 Wh kg$^{-1}$ and 151.9 W kg$^{-1}$, respectively. For the Al/Lig-Cu$_3$Mo$_2$O$_9$@CTAB supercapacitor, the energy and power densities are 28.25 Wh kg$^{-1}$ and 149.43 W kg$^{-1}$, respectively. These values are comparable to the reported works of supercapacitors fabricated with similar materials (Table 1). The EIS test was used to examine the electrical double-layer impedance characteristics of the supercapacitor. A DC voltage = 1 V, AC Voltage = 10 mV, and a frequency between 0.1–106 Hz were applied. Figure 7e,f display the Nyquist plots that were produced for both supercapacitors at zero and the 1200th cycle. A large increase in impedance is observed for the Al/Lig-Cu$_3$Mo$_2$O$_9$@CTAB supercapacitor (Figure 7e). It has an initial impedance of 445 ohms and a final impedance of 4833 ohms. Lower capacity retention occurs due to a reduction in current flow as the system's impedance rises. This results from the charge–discharge process obstructing the ion diffusion channel [47,48]. The impedance performance of the Al/Lig-Cu$_3$Mo$_2$O$_9$@DNA supercapacitor is shown in Figure 7f. It displays a comparatively small increase in impedance even after 1200 cycles. The impedance at zero cycles was 609.2 ohms, and the impedance after 1200 cycles was 1444.4 ohms. The EIS results confirm the retention performance observed earlier. As the system's impedance increases, the current flow decreases, resulting in lower capacity retention.

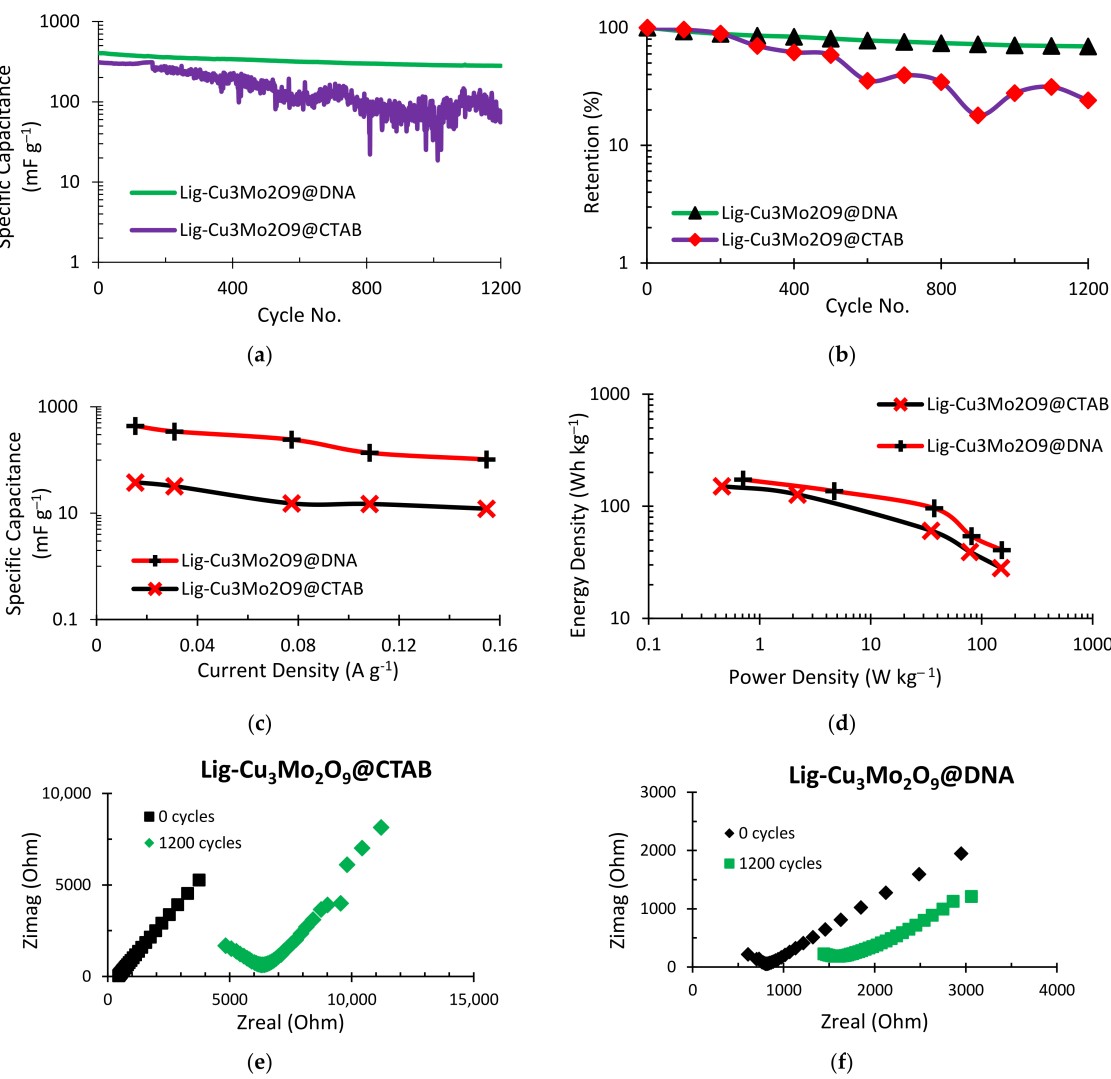

**Figure 7.** CCD plots for (**a**) specific capacitance vs. cycle no.; (**b**) retention (%) vs. cycle no.; (**c**) specific capacitance variation with current density; (**d**) Ragone plot obtained via CCD; (**e**) comparative EIS behavior of Al/Lig-$Cu_3Mo_2O_9$@CTAB supercapacitor; (**f**) comparative EIS behavior of Al/Lig-$Cu_3Mo_2O_9$@DNA supercapacitor.

**Table 1.** Comparison of energy and power densities of the current work with reports in the literature.

| Substrate Material | Electrode Materials | Energy Density (Wh kg$^{-1}$) | Power Density (W kg$^{-1}$) | Current Density | Ref. |
|---|---|---|---|---|---|
| Al | Lignin-$Cu_3Mo_2O_9$@DNA | 40.74 | 151.9 | 0.02 A g$^{-1}$ | This work |
| Al | Lig-$Cu_3Mo_2O_9$@CTAB | 28.25 | 149.4 | 0.02 A g$^{-1}$ | This work |
| Al | Lignin/$MnO_2$ | 6 | 355 | 0.04 A g$^{-1}$ | [12] |
| Al | AC-lignin-$MnO_2$ | 14.11 | 1000 | 6 mA g$^{-1}$ | [6] |
| Al | lig–$NiWO_4$ | 2 | 100 | 0.13 A g$^{-1}$ | [7] |
| Al | lig–$NiCoWO_4$ | 5.75 | 854.76 | 0.8 mA cm$^{-2}$ | [13] |
| - | AC-$CuMoO_4$ | 31.25 | 661.76 | 1.5 mA cm$^{-2}$ | [37] |
| Ni Foam | Carbon Black-$CuMoO_4$ | 10.2 | 500 | 20 A g$^{-1}$ | [49] |
| Ni Foam | rGO/$CuMoO_4$ | 65.6 | 775 | 1.8 A g$^{-1}$ | [50] |
| Ni Foam | 3D Graphene/$CuMoO_4$ | 21.1 | 300 | 1.43 A g$^{-1}$ | [51] |

### 3.6. Role of the Structure-Directing Agents in Formation and Charge-Storage Mechanisms

In order to understand the unstable electrochemical performance of the Al/Lig-$Cu_3Mo_2O_9$@CTAB supercapacitor, we need to look at the unique mechanistic aspects of the formation and storage mechanisms of both nanomaterials. In general, the effects of the morphological confinement of nanomaterials demonstrate an increased surface-to-volume ratio, which results in more favorable capacitive behavior with various 3D transition metal-based materials. Here, the tunability with the morphology was successfully performed with two different structure-directing agents, i.e., an organic compound (CTAB) and the biomolecule (DNA), as a greener alternative. The synthesized $Cu_3Mo_2O_9$ nanomaterials with different structural outcomes can be seen in Figure 3. The morphological results via SEM analysis clearly show the formation of confined nanorods with DNA and 2D nanosheets for CTAB. The formation and storage mechanisms of both the nanorods and nanosheets are shown in Figure 8.

DNA as a biomolecular scaffold plays an important role in the synthesis and formation control of nanomaterials. Initially, the DNA-mediated formation of $Cu_3Mo_2O_9$ involves the functional groups present in the DNA, such as hydroxyl groups, amide groups, sugar moieties, and phosphate groups. In this work, the advantages of the synergism in the presence of such aromatic groups and their heteroatoms presence increases the performance in supercapacitor application. The positively charged metal ions, such as $Cu^{2+}$ and $Mo^{6+}$, electrostatically interact with the above-mentioned negative moieties of DNA and begin to self-assemble over the double-helical structure of DNA (Figure 8a). This mechanism depends on the nucleation of the precursor over the DNA followed by a guided growth via the template shape. This results in the formation of high-surface area $Cu_3Mo_2O_9$ nanorods. The presence of highly dense polymeric chains and the rich functionalities of aromatic base pairs (adenine, guanine, thymine, and cytosine) accommodate the metal ions to form the confined nanorod structures. These confined nanorod structures provide a greater surface area for charge–storage in supercapacitor applications. The structure-directing agent CTAB, which has been widely studied as a successful surface-directing agent for nanomaterial preparation, was selected for improving the wetting ability, emulsification, and interface properties between the electrode and the electrolyte solution. The CTAB-mediated formation of $Cu_3Mo_2O_9$ involves the attraction of $CTA^+$ using the precursor materials via electrostatic forces. The positively charged surfactant with the quaternary ammonium group slows down the formation of the aggregations. As a result, the nanomaterials formed are anisotropic 2D nanosheets (of $Cu_3Mo_2O_9$) (Figure 8a). Comparatively, the formation of $Cu_3Mo_2O_9$ nanorods is quicker and easier to prepare than the surfactant-based nanosheets. DNA is a greener biomolecule; thus, further processing during synthesis is easily avoided. Whereas, when surfactants like CTAB are used, great concerns over the washing and eradicating of the foams after synthesis makes the process tedious.

The electrochemical performance analysis shows that the DNA-mediated synthesis of $Cu_3Mo_2O_9$ (404.64 mF g$^{-1}$) has a higher capacitive performance than the CTAB-mediated $Cu_3Mo_2O_9$ (309.59 mF g$^{-1}$). In order to understand this, the charge storage mechanism in the Lig-$Cu_3Mo_2O_9$@DNA anode and the Lig-$Cu_3Mo_2O_9$@CTAB anode is schematically developed in Figures 8b and 8c, respectively. In the NaOH electrolyte, the Na$^+$ ions are assimilated into the $Cu_3Mo_2O_9$ redox reactions [44,52]. While charging, the Faradaic reaction results in Na$_x$Cu$_3$Mo$_2$O$_9$ formation on the surface of the anode. During the discharge process, the reaction occurs backward, and Na$_x$Cu$_3$Mo$_2$O$_9$ is formed on the surface of the cathode. However, the exact storage mechanism of copper molybdates is still not clear and requires further investigation [35]. The better capacitive performance of Al/Lig-$Cu_3Mo_2O_9$@DNA can be attributed to (i) the interaction of the positively charged metal ions ($Cu^{2+}$ and $Mo^{6+}$) occurring electrostatically with the negative moieties of DNA, which result in efficiently confined, self-assembled nanorods over the double-helical structure of DNA. Thus, reducing the degree of agglomeration during synthesis, which provides more accessible active sites for the electrolyte [53], subsequently improving the electrochemical kinetics of the nanorods when compared to the agglomerated nanosheets. (ii) The nanorod

structure facilitates rapid electron collection pathways and serves as ion storage for rapid ionic diffusion, thereby boosting the use of active materials [52].

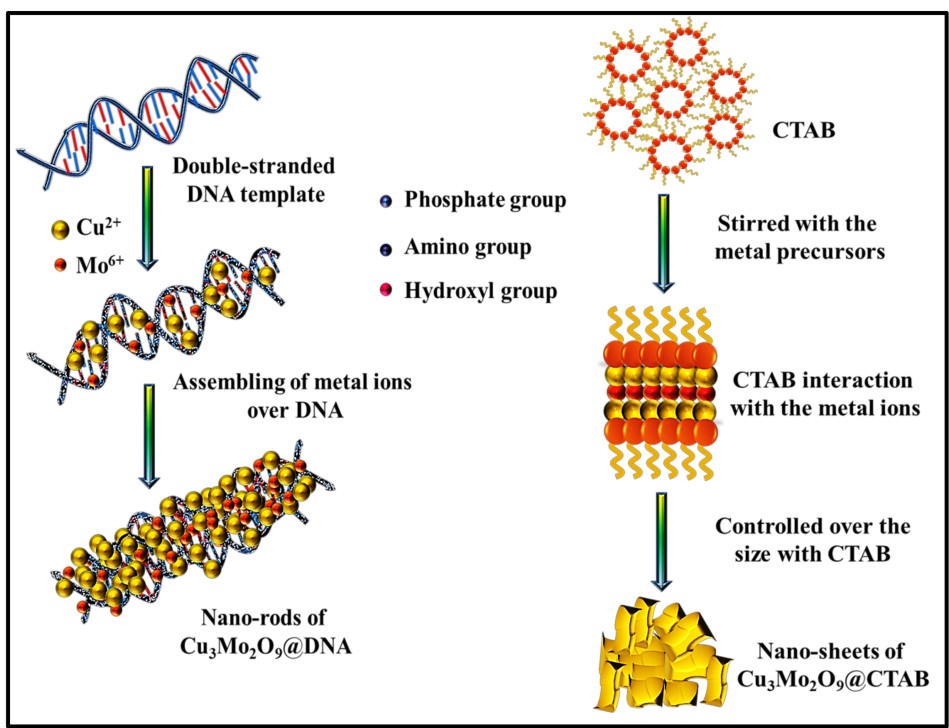

(**a**)

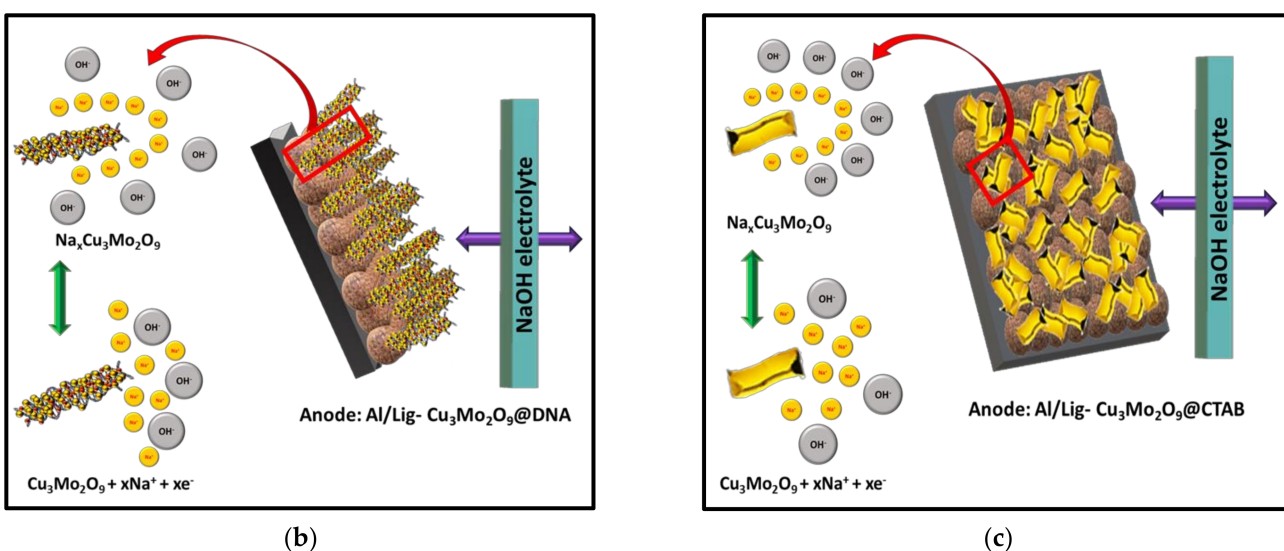

(**b**)          (**c**)

**Figure 8.** (**a**) Formation mechanism of $Cu_3Mo_2O_9$ using DNA and CTAB; storage mechanism during charging for (**b**) $Cu_3Mo_2O_9$@DNA nanorods in the Al/Lig-$Cu_3Mo_2O_9$@DNA supercapacitor; (**c**) $Cu_3Mo_2O_9$@CTAB nanosheets in the Al/Lig-$Cu_3Mo_2O_9$@CTAB supercapacitor.

## 4. Conclusions

In this research, $Cu_3Mo_2O_9$ nanoparticles were successfully synthesized via two different fabrication methods. The $Cu_3Mo_2O_9$ nanoparticles were developed using CTAB and DNA. The morphological analysis of these nanoparticles depicts the nanorod structures obtained via the DNA method and the perfect nanosheets obtained via the CTAB method. Upon the synergistic integration of the nanoparticles into the lignin matrix, the electrochemical properties were enhanced. A higher capacitive performance for the Al/Lig-$Cu_3Mo_2O_9$@DNA supercapacitor was observed when compared to the Al/Lig-

Cu$_3$Mo$_2$O$_9$@CTAB supercapacitor. The Al/Lig-Cu$_3$Mo$_2$O$_9$@DNA supercapacitor had a specific capacitance value of 404.64 mF g$^{-1}$ with a ~70% retention, and energy and power density values of 40.74 Wh kg$^{-1}$ and 151.9 W kg$^{-1}$, respectively. The unique architecture and uniform surface morphology of the Cu$_3$Mo$_2$O$_9$@DNA nanoparticles boosts the charge–storage performance. Consequently, the effective fabrication of high-performing, solid-state supercapacitors from biomaterials with shorter synthesis times was successfully achieved. This research offers a straightforward and inexpensive method for creating Cu$_3$Mo$_2$O$_9$ composite electrodes based on lignin for supercapacitor applications. The flexibility of the supercapacitors in this work is imparted by the Al foil substrate, with a thickness of 0.98 mm, used to fabricate them. The small thickness of the Al foil makes them flexible in nature. This work focuses on the electrochemical performance of the supercapacitors as they are: flat without any induced mechanical properties. In order to understand the effect of the mechanical properties of flexible supercapacitors on their capacitive performance, further studies, including bending and tensile tests, need to be performed in the future.

**Supplementary Materials:** The following supporting information can be downloaded at: https://www.mdpi.com/article/10.3390/jcs7040155/s1. Figure S1. Highly magnified XPS spectra of individual elements N 1s and C 1s for the precursor (a–b) DNA; (c–d) CTAB.

**Author Contributions:** Conceptualization: S.M., S.J. and H.L.; Methodology: S.M., S.K. (Sangeetha Kumaravel) and S.J.; Validation: S.M. and S.K. (Sangeetha Kumaravel); Formal Analysis: S.M.; Investigation: S.K. (Sangeetha Kumaravel) and M.Y.; Resources: S.K. (Subrata Kundu) and H.L.; Data Curation: S.M. and M.Y.; Writing—Original Draft Preparation: S.M.; Writing—Review and Editing: S.M., S.K. (Sangeetha Kumaravel), S.K. (Subrata Kundu) and H.L.; Visualization: S.M. and S.K. (Sangeetha Kumaravel); Supervision: S.K. (Subrata Kundu) and H.L.; Project Administration: H.L. All authors have read and agreed to the published version of the manuscript.

**Funding:** This research received no external funding.

**Data Availability Statement:** The data presented in this study are contained within the article and/or Supplementary Materials.

**Acknowledgments:** The authors would like to thank Alex Fang at ETID for his help with critical discussion and data collection. The Aggie Research Program made the participation of undergraduate students in this research possible.

**Conflicts of Interest:** The authors declare no conflict of interest.

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
