# Peer review of "Impacts of Structure-Directing Agents on the Synthesis of Cu3Mo2O9 for Flexible Lignin-Based Supercapacitor Electrodes"

_jcs, doi:10.3390/jcs7040155_

Round 1
Reviewer 1 Report
Cu3Mo2O9 nanomaterials with different morphologies were prepared by using the structure directing agent with negative and positive charges. The manuscript was well organized and prepared. However, the manuscript still existed some problems, therefore, it should undergo major revisions before its acceptance.
(1) Page 2/15: //The exceptional specific surface area, active sites, porosity, and electrical conductivity of lignin nanoparticles make them especially promising for developing green energy storage systems.// However, lignin is a poor conductor, so lignin nanoparticles should be revised to lignin-based carbon nanoparticles or lignin-derived carbonaceous nanoparticles.
(2) Why the annealing temperature for preparing Cu3Mo2O9@DNA was 600℃, while the annealing temperature for preparing Cu3Mo2O9@CTAB was 550℃? In fact, the annealing temperature will influence the structure and properties of the prepared samples.
(3) There are many kinds of structure-orienting agents. Therefore, the reasons for selecting CTAB and DNA as the structure-directing agents for preparing Cu3Mo2O9 should be briefly clarified. Is it because DNA has negatively charged groups and CTAB has positively charged groups?
(4) Were the structures and properties of the biomaterial activated carbon (AC) from Sigma-Aldrich the same? If not, the manuscript should also provide the specific surface area and pore structure parameters of this AC.
(5) Title: //Impact of Structure-Directing Agents in Synthesis of Cu3Mo2O9 for Flexible Supercapacitor Electrodes//. The flexibility of supercapacitor electrode was derived from lignin. Therefore, it is suggested to integrate "lignin" into the Title of the manuscript.
(6) It is suggested to expand the range of Raman shift in Raman spectrum measurement to confirm whether there are Raman peaks corresponding to the D band and G band of carbon in the Raman spectra, and to indicate the decomposition or partial carbonization of the structure directing agents (DNA and CTAB) after annealing. In fact, carbon was clearly present in the EDS spectra in Figure 4.
(7) //From the low-magnification SEM images of Cu3Mo2O9@DNA (Figure 3a), a nanorod structure can be observed with 300-400 nm length and ~400 nm diameter.// But from the SEM in Fig. 3b, its length was much longer than its diameter.
(8) When alkaline electrolyte is used, KOH solution is usually selected. So this manuscript should briefly explain why NaOH was used as electrolyte.
(9) For higher scan rates, it was seen that the rate of diffusion increased while the specific capacitance decreased. Herein, what did “rate of diffusion” mean?
(10) Page 9/15: “Al/Lig- Al/Lig-Cu3Mo2O9@CTAB supercapacitor” should be revised.
(11) //The Al/Lig-Cu3Mo2O9@DNA supercapacitor exhibits a greater specific capacitance during the CV testing. This can be attributed to the two different synthesis methods used to fabricate the Cu3Mo2O9 nanoparticles.// The above statement did not really explain the reason why the Al/Lig-Cu3Mo2O9@DNA supercapacitor showed a greater specific capacitance. Therefore, it is suggested that the explanation can be made based on the electrostatic assembly of positively charged metal ions (M+) and negatively charged DNA groups, like Line 348-352.
(12) //For the Al/Lig-Cu3Mo2O9@DNA supercapacitor, the energy and power densities measured at current density of 20 mA g-1 (0.02 A g-1) are 40.74 mWh kg-1 and 151,950.46 mW kg-1, respectively. For the Al/Lig-Cu3Mo2O9@CTAB supercapacitor, the energy and power densities are 28.25 mWh kg-1 and 149,427.46mW kg-1, respectively.// Compared with the following works, as far as I know, J. Mater. Chem. A, 2022, 10, 9837-9847; ACS Appl. Mater. Interfaces, 2022, 14, 33328−33339, the above energy density was very low. Therefore, it is recommended to compare the energy density of the supercapacitor in this manuscript with those of supercapacitors assembled with other similar materials so as to show the performance level of the supercapacitor of this manuscript.
(13) The newest review articles on supercapacitor/hybrid energy storage, Chinese Chem. Lett., 2023, 34, 107784; Chem. Eur. J., 2023, 29, e202203973; J. Mater. Chem. A, 2023, DOI: 10.1039/D2TA09258A, are recommended to be included in the References for balanced citations, providing more valuable information for the broader audience.
(14) The effect of mechanical properties of flexible supercapacitors on their specific capacitance and energy density should be supplemented.
(15) The update References from 2022-2023 should be supplemented.
Author Response
Dear Reviewer,
Thank you for giving us the opportunity to revise the manuscript “Impact of Structure-Directing Agents in Synthesis of Cu3Mo2O9 for Flexible Supercapacitor Electrodes”. We greatly appreciate your and reviewers’ time spent on this. We have gone through a thorough revision of the work accordingly. The feedback helped us improve the paper to a great extent. The changes made to the original version were tracked and are being submitted.
Please feel free to contact us should more information be needed.
Sincerely,
Hong Liang
Oscar S. Wyatt Jr. Professor
Texas A&M University, College Station, TX

Reviewer 2 Report
The author used two different agents to control the morphology of electrochemical active Cu3Mo2O9. The materials have been well analyzed, the effect of morphology on electrochemistry is also investigated. The manuscript is well written, and the presented results validate the conclusions. I suggest this work can be accepted after a minor revision to address the following questions.
1. Please confirm the chemical formula of the material. It is written as Cu2Mo3O9 in the title, but from the main text it should be Cu3Mo2O9.
2. Why did the authors use mF cm-2 as the unit to calculate the capacitance? The mA g-1 was used for the current density, mW kg-1 for the power density and mWh kg-1 for the energy density, therefore a weight specific capacitance would be more reasonable. The theoretical capacitance should be given for a reference.
3. The authors should give more explanation about why the y axis of cycling stability performance is exponential, meanwhile why Cu3Mo2O9@CTAB sample show very unstable performance.
Author Response

(The authors gave the same response as above.)

Round 2
Reviewer 1 Report
Accept in present form.
Reviewer 2 Report
The authors have addressed the questions and comments properly, not only from my side but also from other reviewer. I would recommend to accept the manuscirpt in the present form.